# Effect of Soil Moisture Regimes on the Glyphosate Sensitivity and Morpho-Physiological Traits of Windmill Grass (*Chloris truncata* R.Br.), Common Sowthistle (*Sonchus oleraceus* L.), and Flaxleaf Fleabane [*Conyza bonariensis* (L.) Cronq.]

**DOI:** 10.3390/plants10112345

**Published:** 2021-10-29

**Authors:** Arslan Masood Peerzada, Alwyn Williams, Chris O’Donnell, Steve Adkins

**Affiliations:** School of Agriculture and Food Sciences, The University of Queensland, Gatton, QLD 4343, Australia; alwyn.williams@uq.edu.au (A.W.); c.odonnell@uq.edu.au (C.O.); s.adkins@uq.edu.au (S.A.)

**Keywords:** climate-induced herbicide tolerance, northern grain region, herbicide sensitivity, summer fallow, climatic stress

## Abstract

The glasshouse study was conducted with the objectives of (i) investigating the effect of soil moisture variations on the control efficiency of glyphosate on windmill grass (*Chloris truncata* R.Br.), common sowthistle (*Sonchus oleraceus* L.), and flaxleaf fleabane [*Conyza bonariensis* (L.) Cronq.], (ii) evaluating the tolerance of tested weed species under soil moisture variations, and (iii) determining the morphological and physiological characteristics of these species to partially explain herbicide tolerance under periods of reduced soil moisture availability (RSM). The species’ tolerance to glyphosate increased significantly under reduced soil moisture availability (*p* < 0.001). The lethal dose to cause herbicide injury or biomass reduction by 50% (LD_50_) and 80% (LD_80_) in relation to untreated control for water-stressed plants [i.e., moderate soil moisture availability (MSM) and RSM] was significantly higher than that of plants grown under high soil moisture availability (HSM). The tolerance factor (TF) for *C. truncata*, *S. oleraceus*, and *C. bonariensis*, in terms of biomass reduction under RSM, was 2.6, 2.4, and 2.6, respectively, as compared to HSM. The results showed that the glyphosate sensitivity, especially at the sub-lethal rates, of the three weed species under study decreased as soil moisture availability reduced (*p*
*<* 0.01). Overall glyphosate efficacy, in relation to the recommended rate, was unaffected, except for *C. truncata*; the weed survived the highest tested glyphosate rate [750 g active ingredient (a.i.) ha^−1^] under RSM. There was significant interaction between weed species and soil moisture regimes for weed morpho-physiological traits (*p* < 0.001), with reduced soil moisture having a more influential impact on the growth of *C. bonariensis* and *S. oleraceus* compared to *C. truncata*. Changes in the leaf characteristics, such as increased leaf thickness, higher leaf chlorophyll content, reduced leaf area, and limited stomatal activity for all the tested weed species under MSM and RSM in relation to HSM, partially explain the tolerance of species to glyphosate at sublethal rates.

## 1. Introduction

Windmill grass (*Chloris truncata* R.Br.), common sowthistle (*Sonchus oleraceus* L.), and flaxleaf fleabane [*Conyza bonariensis* (L.) Cronq.] are the most important weed species of Australia’s northern grain region (NGR), mainly due to their prolific production of highly dispersive seeds, low innate seed dormancy, prolonged emergence period, rapid plant maturity, and high risk of evolving glyphosate resistance [1]. These ephemeral, summer-growing weed species germinate following episodic summer rain and grow on summer fallows, transpiring water that could otherwise be stored in the soil for use by subsequent cash crops [2]. Additionally, the primary source of nitrogen (N) for subsequent cash crops comes from the mineralization of organic matter following summer rainfall events [3]. Thus, weed infestations during the summer fallows reduce mineral N availability, either by drying the soil or by accumulating N in weed biomass, which negatively impacts subsequent crops by reducing grain yield and protein content. Therefore, the management of summer fallow weeds is critical since the cost of controlling such weeds will be much lower than if they are left unmanaged [4].

Glyphosate (N-[phosphonomethyl]-glycine) has become the most widely used non-selective herbicide since its commercialisation by Monsanto in the 1970s [5]. Being a systemic herbicide, it disrupts the synthesis of aromatic acids via the shikimic acid pathway by competitive inhibition of the chloroplast localised enzymes, EPSP (5-enol pyruvyl-shikimate-3-phosphate) synthase [6]. Due to its desirable characteristics, such as high efficacy, broad spectrum, low cost, and a good toxicological profile, combined with the introduction of genetically modified herbicide-tolerant crops, glyphosate is widely considered to be the most important herbicide for summer fallow weed management [4,7]. However, glyphosate has been used with variable levels of success in the northern grain region (NGR) of Australia [8,9,10,11,12].

Practical experience shows that glyphosate’s success as a broad-based herbicide, resting on the promise of ever-increasing farm efficiency and productivity, relies not only on its chemical properties but also on its interaction with the plant and the environment [8,11]. Besides plant morpho-physiologic, anatomic, and molecular characteristics, environmental conditions before, during, or after herbicide application play a pivotal role in determining the efficacy and performance of glyphosate [13]. In the regions where summer temperatures are projected to increase, rainfall deficiencies resulting from high variability often result in water-deficit stresses, which severely influence plant growth, development, and survival depending on the severity and duration of stress, plant species, and the stage of growth at which stress occurs [14,15,16].

Changes in soil water potential affect weed growth and development and may also affect herbicide efficacy [17,18]. Numerous greenhouse and growth chamber studies have demonstrated that reduced soil moisture reduces the effectiveness of glyphosate on barnyard grass (*Echinochloa crus-galli* (L.) Beauv.], common milkweed (*Asclepias syriaca* L.), Johnsongrass [*Sorghum halepense* (L.) Pers.], purple nutsedge (*Cyperus rotundus* L.), wild oat (*Avena fatua* L.), liverseed grass (*Urochloa panicoides* Beauv.), and awnless barnyard grass [*Echinochloa colona* (L.) Link] [11,12,19,20,21,22,23]. The possible mechanism associated with the reduced activity of glyphosate is related to alterations in herbicide retention, absorption, translocation, and/or metabolism due to changes in leaf thickness, leaf area, and stomatal conductance [20,22,23]. These morphological and physiological changes in the leaf architecture and surfaces, which alter herbicide absorption, translocation, and metabolism, reduce the efficacy of glyphosate under moisture-deficit stress.

Regional projections suggest that south-eastern Australia will be drastically affected by changes in rainfall patterns, as well as rising temperatures, which increase the severity of the drought [24]. The highest level of genetic diversity and the ability to withstand drought will allow some weeds to escape chemical management successfully or partially and, thus, adapt, establish, and spread across the agroecosystems [25,26]. Though soil moisture directly influences weed physiology, some species have a higher tolerance than other species [27], depending on the relationship between selection pressure and gene-flow. Therefore, the differences in response to water stress due to their alternation in various phenological and physiological processes, which vary with the degree and length of water stress [28], could influence herbicide tolerance, either positively or negatively.

Based on the above-mentioned literature, it was hypothesized that glyphosate efficacy would decrease under drought conditions as weed tolerance to herbicide would increase under severe moisture-deficit stress [29,30], but the impact would be species- and rate-specific. The objectives of the present study were (i) to quantify the growth and physiological responses of three problematic weed species (*C. truncata*, *S. oleraceus*, and *C. bonariensis*) of NGR of Australia when grown under different soil moisture regimes, (ii) examine the effect of different soil moisture regimes on the efficacy of glyphosate when applied to these three problematic weed species, and (iii) to determine which growth and physiological responses may be responsible for the water stress-induced tolerance to glyphosate.

## 2. Results

### 2.1. Impact of Soil Moisture Variations on Weed Susceptibility to Glyphosate

Glyphosate application resulted in a significant reduction in the leaf chlorophyll content of *C. truncata*, *S. oleraceus*, and *C. bonariensis* (*p* < 0.001; Figure 1) when compared to non-treated controls under high (HSM) and moderate soil moisture availability (MSM). However, a gradual increase in leaf chlorophyll content was observed for all the tested weeds at 4 DAA when glyphosate was applied at sublethal rates (i.e., 62.5 to 250 g a.i. ha^−1^) under reduced soil moisture availability (RSM), as shown in Figure 1. In the case of *C. truncata*, the reduction of leaf chlorophyll content, over time (0 to 6 DAA) under HSM (81 to 95%) and MSM (43 to 75%) was higher as compared to RSM (25 to 38%) when sprayed above 375 g a.i. ha^−1^ (*p* < 0.001; Figure 1).

In comparison to SPAD values at 0 DAA, the leaf chlorophyll content for *S. oleraceus* under HSM decreased by 66 to 80% at 6 DAA when glyphosate was applied at >375, whereas the values ranged between 31 and 52% and 24 and 39% under MSM and RSM, respectively (*p* < 0.001; Figure 1). An increased level of tolerance was observed in *C. bonariensis*; plants treated with glyphosate at a rate between 125 and 375 g a.i. ha^−1^ showed a reduction of leaf chlorophyll content by 2 to 48%, whereas the reduction ranged between −10.5 and 21% and −4 and 5.6% in MSM and RSM, respectively, at similar rates, in relation to controls at 0 DAA (*p* < 0.001; Figure 1). The results thus confirm differential herbicide sensitivity of tested weed species under soil moisture variations during the first week.

The visual herbicide injury (%) and biomass reduction (%) for tested weed species as a function of glyphosate rates and soil water regime, fitted to sigmoidal regression models in Eqn. 1 and Eqn. 2 (*p* < 0.001; Figure 2; Table 1). The values of the determination coefficient (R^2^) varied from 0.81 to 0.98, demonstrating the satisfactory adjustment of data to the models (Table 1). For all the tested species, a significant effect (*p* < 0.05) of soil water content and herbicide rates could be observed on glyphosate susceptibility. With the lack of overlap of the confidence interval (CI) of the species under well-watered conditions (HSM) in relation to moderate (MSM) and severe water stress (RSM), it was possible to establish the tolerance factor (TF) for both herbicide injury (%) and biomass reduction (%) using Equations (2) and (3).

The results showed that higher LD_50_ values (316.7 g a.i. ha^−1^) of glyphosate was required to cause 50% injury to *C. truncata* under RSM in comparison to other soil water contents; glyphosate at 122.2 and 244.5 g a.i. ha^−1^ caused 50% injury under HSM and MSM, respectively (Figure 2; Table 1). For dry biomass reduction, a similar pattern was observed where a 50% reduction was recorded with glyphosate at 239.4 g a.i. ha^−1^ at RSM, which is 2 to 2.6 times higher than other soil moisture contents. Thus, the results indicate that *C. truncata* was 2 to 3-fold more tolerant under MSM and RSM as compared to HSM. The regeneration of lateral shoots was also observed, particularly under MSM and RSM.

Similarly, the LD_50_ values estimated for the herbicide injury and biomass reduction for *S. oleraceus* and *C. bonariensis* deviated significantly between the soil moisture variations. Glyphosate at 120.7, 225.3, and 290.4 g a.i. ha^−1^ caused 50% injury to *S. oleraceus* under HSM, MSM, and RSM, respectively (Figure 2; Table 1). The LD_50_ value for biomass reduction under HSM was 110.1 g a.i. ha^−1^, whereas the value under MSM and RSM was 178.9 and 264.9 g a.i. ha^−1^, respectively. The tolerance factor, based on biomass reduction, at RSM and MSM was 2.4 and 1.6 in comparison to HSM (Table 1). For *C. bonariensis*, it was determined that 250.9 and 317.6 g a.i. ha^−1^ were required to cause 50% injury under MSM and RSM, respectively, and for HSM, it was 150.9 g a.i. ha^−1^ (Figure 2; Table 1). In terms of biomass reduction, plants showed the highest tolerance to glyphosate in RSM, with an LD_50_ of 347.2 g a.i. ha^−1^, followed by MSM and HSM, with LD_50_ of 211.6 and 131.1 g a.i. ha^−1^, respectively. The TF values were 1.6 and 2.6 at MSM and RSM, respectively, showing that the rate required to cause 50% biomass reduction was two times greater than the one required under HSM (Table 1).

Although differences in the glyphosate rate to cause 80% injury or biomass reduction were observed for all tested weed species under soil water variations, results showed that glyphosate within the recommended rates provided satisfactory control of tested weed species (Table 1). For example, glyphosate rates to cause 80% reduction of dry biomass in *C. truncata*, *S. oleraceus*, and *C. bonariensis* under HSM were 236.1, 234.6, and 289.0 g a.i. ha^−1^, respectively. However, the LD_80_ values under RSM were 506, 340.6, and 474.2 g a.i. ha^−1^, which is within the range of glyphosate rates recommended for *C. truncata* (740 g a.i. ha^−1^), *S. oleraceus* (705 g a.i. ha^−1^), and *C. bonariensis* (658 g a.i. ha^−1^) in the NGR, respectively. Even with such low levels of tolerance, there are indications that there will be increased chances of tolerance to glyphosate under MSM and RSM, especially at the sublethal herbicide rates, which might vary with natural tolerance of the weed species.

### 2.2. Impact of Soil Moisture Variations on Weed Morpho-Physiology and Growth

For all the tested weed species, the plant height attained, the number of leaves produced, their total leaf area, and above-ground plant dry biomass all decreased when exposed to the degree of water stress (*p* < 0.001; Table 2, Table 3 and Table 4). For *C. truncata*, the plant height at MSM and RSM decreased by 26% and 46%, respectively, as compared to HSM. The number of leaves produced decreased by 24% at MSM and 50% at RSM as compared to HSM. The leaf area was significantly higher for HSM (267.1 cm^2^), followed by MSM (174.2 cm^2^) and RSM (129.7 cm^2^). Compared with HSM, the dry biomass of plants was reduced by 33% and 58% at MSM and RSM, respectively. However, the leaf chlorophyll content and leaf thickness all increased when exposed to soil moisture stress conditions (Table 2). The chlorophyll content at MSM and RSM increased by 28% and 64%, respectively, as compared to HSM. Leaf thickness at MSM and RSM increased by 10% and 20%, respectively, as compared to HSM. Stomatal conductance at MSM and RSM decreased by 22% and 51%, respectively, as compared to HSM.

The response of *S. oleraceus* was similar: water-stress treatments negatively affected nearly all parameters measured at 55–60 days after emergence (*p* < 0.001; Table 3). Plant height decreased by 23% at MSM and 68% at RSM as compared to HSM. The number of leaves produced at MSM and RSM decreased by 35 and 47%, respectively, at MSM and RSM as compared to HSM. The chlorophyll content of *S. oleraceus* leaves was significantly higher at MSM (17%) and RSM (31.1%) as compared to HSM. However, stomatal conductance decreased as stress intensified; the value decreased by approximately 27% and 63% as WHC decreased from HSM to MSM and RSM, respectively. The total leaf area, dependent on plant growth and the number of leaves, was recorded at the highest with HSM, whereas the value decreased by 42% and 61% with increasing water stress. Relative to the HSM, the thickness of the leaf at the laminar portion increased substantially by 6% and 20% with MSM and RSM, respectively. The highest dry biomass was recorded in plants growing at HSM (3.1 g), whereas it was reduced by 32 and 61% when maintained at MSM and RSM, respectively.

The morpho-physiological and growth responses of *C. bonariensis* were more severely impacted by water stress as compared to *C. truncata* and *S. oleraceus* (*p* < 0.001; Table 4). Plant height decreased by 58% and 68% with MSM and RSM, respectively, as compared to HSM. The number of leaves produced at MSM and RSM decreased by 43 to 73% relative to HSM. The ANOVA showed that the amount of leaf chlorophyll content increased by 14.2% and 29.1% under MSM and RSM as compared to HSM. A better water supply (HSM) resulted in significantly higher stomatal conductance, while MSW to RSM reduced stomatal conductance by 47.6–73.3%. Plants subjected to MSM and RSM showed an increased leaf thickness of 14.5% and 26.5%, respectively, which differed significantly from HSM. The total leaf area reached its maximum at HSM (250.0 cm^2^), whereas the value reduced as the degree of water stress increased (i.e., 160.9 cm^2^ at MSM and 87.7 cm^2^ at RSM). Plant dry weight decreased in both MSM and RSM by 25% and 67.9%, respectively, as compared to HSM.

## 3. Discussion

### 3.1. Impact of Soil Moisture Variations on Weed Susceptibility to Glyphosate

Several studies have highlighted the influence of limited moisture availability on the glyphosate sensitivity of weed species, but the information available on the response of *C. truncata*, *S. oleraceus*, and *C. bonariensis* is scarce. Glyphosate efficacy decreased as soil moisture decreased, as evidenced by leaf chlorophyll reduction (%), herbicide injury (%) and biomass reduction (%) for all three weed species. These results support that soil moisture content plays a critical role in predicting the efficacy of glyphosate on these weed species, particularly on *C. truncata* and *C. bonariensis*. These results also support the assertion that the difference in LD_50_ and LD_90_ values between stressed and non-stressed plants is due to a difference in the plant-herbicide-moisture interactions.

Studies have reported SPAD readings as a good predictor of leaf chlorophyll content [31]; few studies have used this method in assessing glyphosate activity [32]. Analysis of variance related to the main effect of glyphosate and weed species indicated significant differences in leaf chlorophyll content across soil moisture regimes. The leaf chlorophyll in *C. truncata*, *S. oleraceus*, and *C. bonariensis* content declined significantly following application of glyphosate under HSM (Figure 1). However, moderate to a minimum reduction in leaf chlorophyll content values were recorded under MSM and RSM, respectively. Our findings were at par with the findings of Tanpipat [32]. The results indicated that glyphosate within recommended rates under HSM and MSM was effective and consistent in causing a significant reduction in leaf chlorophyll content, while all the tested weed species showed increased tolerance when grown under RSM, with *C. bonariensis* showing the highest values. The increased herbicide tolerance under RSM could possibly be due to the presence of trichomes which restrict or delay the penetration and absorption of glyphosate [33].

Despite slight to moderate differences at sub-lethal rates, glyphosate application at the recommended rate under HSM caused the highest herbicide injury and resulted in the lowest weed growth as indicated by poor dry biomass accumulation for all the three tested weed species (Table 1; Figure 1 and Figure 2). However, under RSM, this response to glyphosate was species-specific, as *C. truncata* tolerated glyphosate more efficiently than *S. oleraceus* and *C. bonariensis*, resulting in 10% more plant survival and more biomass at the highest tested rate. Plants of *C. truncata* at sub-lethal rates lost apical dominance and initiated multiple lateral buds when grown under MSM and RSM. In addition, glyphosate at 62.5 and 125 g a.i. ha^−1^ stimulated seedling growth under MSM and RSM, possibly due to its hormetic effect [34].

Taken across all three soil water conditions, herbicide activity shows a correlation with the soil moisture regime, as has been demonstrated for several post-emergent herbicides [13,35]. Though species’ tolerance to glyphosate increased two to three-folds under the water-stressed conditions relative to the well-watered condition, glyphosate application within the recommended rates provided complete control of all the tested weed species. These results were in line with the finding of Adkins et al. [11], Adkins et al. [12] and Tanpipat [32] on *U. panicoides*, *E. colona*, and *A. fatua* that a reduced soil moisture content resulted in increased species’ tolerance and the time required to control the same population under optimal water conditions. Similar observations for glyphosate were also reported in other species, such as black nightshade (*Solanum nigrum* L.), cogon grass [*Imperata cylindrical* (L.) Beau.], and purple nutsedge [*Cyperus rotundus* (L.) Pers.] [22,36].

Though the study indicated that the tested weed species differ in their sensitivity to glyphosate rates under varying soil moisture conditions, the fundamental reasons behind the increased herbicide tolerance at RSM reduced soil water content are not known; studies on herbicide absorption, translocation, and metabolism under water-deficit stress could explain the phenomenon behind increased tolerance. With respect to glyphosate, nothing has been reported previously to explain its reduced activity in *C. truncata*, *S. oleraceus*, and *C. bonariensis* when applied under moisture stress. One further possibility of increased water stress-induced tolerance in the present study could be that the generated water stress altered the morpho-physiology, and this may lead to reduced glyphosate interception, penetration, absorption, and translocation. These changes in physiological as well as morphological responses to reduced soil moisture conditions or water stress are reported for the relative inefficiency of glyphosate against weeds under extreme soil moisture stress [22]. These morpho-physiological adaptive adjustments in response to reduced soil water content could enable weed species to avoid or escape glyphosate treatment. Possibilities exist that glyphosate at recommended rates might act as sublethal rates, depending upon the species’ natural tolerance, when sprayed in periods of drought, which might help weeds in resistant evolution over multiple generations [37].

### 3.2. Impact of Soil Moisture Variations on Weed Morpho-Physiology and Growth

The objective of this study was to quantify the growth response under varying soil moisture regimes in *C. truncata*, *S. oleraceus*, and *C. bonariensis*. For all the tested weed species, highly significant differences in growth parameters were recorded across varying soil moisture regimes (Table 2, Table 3 and Table 4). The reduction in the availability of water imposed structural changes on the leaves and affected their physiological performance. Leaf structural and physiological traits varied significantly in response to water regimes in all the species. The present data is similar to that seen previously where water stress has been shown to reduce plant height, leaf area, plant biomass, and root:shoot ratio [38,39]. In addition, other studies have shown water stress to affect many plant processes, such as photosynthesis, protein synthesis, and the build-up of metabolites [40,41]. Such changes may explain the substantial decrease in the leaf number and area, and, in turn, in plant biomass seen in our study. The reduction in leaf mass and area could be from producing smaller, thicker leaves and reducing stomatal conductance, as observed in our study, to minimize water loss through epidermal cells [42].

Previous studies have reported a substantial reduction in growth, physiology, and seed production of several weed species, including barnyard grass [*Echinochloa crus-galli* (L.) Beauv.], velvetleaf (*Abutilon theophrasti* Medicus), black knapweed (*Centaurea nigra* L.), and pigweed (*Amaranthus retroflexus* L.) under varying degrees and durations of water stress [42,43,44,45]. A recent study on *C. truncata* also reported that the productivity of the plants, as measured by the dry biomass per plant, was reduced with moisture stress [46]. These studies reported that water stress affects most growth functions, but these effects depend on the level of water stress, the length of time the plants are subjected to it, and the plant species [47].

Though the herbicide within the recommended rate provided complete control of the tested weed species, increased tolerance to sub-lethal rates could be crucial for understanding the eco-evolutionary insight of climate-induced herbicide tolerance and the slow evolution of herbicide resistance. The morpho-physiological and growth features can partially explain the variations between the species and water-deficit stress against herbicides, especially at sublethal rates. For example, plants of *C. truncata*, *S. oleraceus*, and *C. bonariensis* when exposed to water stress showed certain symptoms, such as reduced plant height and leaf surface area, altered leaf position or angle, and stunted growth or reduced biomass production, and leaf folding (i.e., *C. truncata*). These morphological features related to leaf surface characteristics, especially leaf architecture (e.g., leaf area and angle), as well as height, in relation to the spray nozzle, directly influence the amount of herbicide spray droplet deposition and retention on the leaf surface [47,48,49]. In addition, water-deficit conditions increase leaf cuticle thickness, as observed in our study, alter the epicuticular wax composition, and increase total wax content [50], which can inhibit the absorption and penetration of post-emergent herbicides, such as glyphosate [51].

With the onset of water-deficit stress, the stomatal conductance of matured and fully-developed leaves declined in *C. truncata*, *S. oleraceus*, and *C. bonariensis* to restrict water transpiration, in order to avoid dehydration [52]. Thus, decreased stomatal conductance or the number of stomata reduces the penetration of glyphosate into leaves, which also reduces the efficacy of glyphosate [53]. As the stomatal conductance is reduced, due to decreased xylem water potential, photosynthesis and transport of photosynthates out of the leaf decreases [54,55], which restricts the translocation of phloem-mobile herbicides like glyphosate.

There is also indirect evidence that chlorophyll is jointly controlled by climate and soil. Thus, chlorophyll might be an indicative trait for characterising how plants respond to climate change. Earlier studies have found that photosynthesis is not the primary inhibitory target of glyphosate, and glyphosate injury is found to be correlated with chlorophyll content [56,57]. In our study, the content of chlorophyll in the leaves of tested species increased significantly in response to water stress, which is the most interesting trait. Tolerance to glyphosate under MSM and RSM, especially at sublethal rates, could possibly be related to the higher leaf chlorophyll content, which has not been investigated before. This can be justified by the findings of Darwish et al. [58], who reported a differential response of two tobacco varieties to clomazone, which inhibits biosynthesis of photosynthetic pigments, because both chlorophyll and carotenoids were two times higher in *Virginie* compared to *Xanthi*. Possibilities exist that a higher leaf chlorophyll content under water stress will increase the number of target-sites (EPSPS), and glyphosate molecules at the recommended rate might act as sublethal, thus requiring further investigation.

## 4. Materials and Methods

### 4.1. Seed Collection

The seeds of windmill grass (*Chloris truncata* R.Br.), common sowthistle (*Sonchus oleraceus* L.), and flaxleaf fleabane [*Conyza bonariensis* (L.) Cronq] were collected during 2017/2018 from southeast and central Queensland (QLD) with locations being recorded by geographic positioning systems (GPS). These species were selected based on previous reports of unsatisfactory control with glyphosate, their invasiveness and impact characteristics, their current and potential area of spread, and their environmental and socioeconomic impacts on Australia’s NGR. Three or four biotypes of each species were collected to examine intraspecific variation across a large geographic area. The seeds of each collected biotype were dried in the shade for ~7–10 days, separated from the chaff, mixed thoroughly, and stored under paper bags in a seed storage room (15 ± 1 °C; 15 ± 3% relative humidity) until used.

### 4.2. Dose-Response Study to Determine LD_50_ and LD_80_ Values

A preliminary rate-response study was conducted to identify the glyphosate susceptibility levels for each biotype of the weed species. Based on the results, the highly susceptible biotype for each weed species was used in the water-stress experiment. The LD_50_ recorded for the highly susceptible biotypes of *C. truncata*, *S. oleraceus*, and *C. bonariensis*, based on biomass reduction, were 80.0, 79.3, and 130.2 g a.i. ha^−1^, respectively, when grown in a glasshouse with a mean temperature of 30/20 ± 2 °C, a photoperiod of 12/12 h (day/night), and relative humidity of 65 ± 5% (data not presented).

### 4.3. Determination of Water Holding Capacity

For both the experiments, seedlings of tested weed species were grown at 3 different soil moisture regimes: 90–100%, 50–60%, and 20–30% of the water holding capacity (WHC), determined by the soil gravimetric method of Gardner [59], modified by Adkins et al. [12] and Tanpipat et al. [8]. The procedure has been discussed in detail by Peerzada [60] and Bajwa et al. [25]. Based on that calculation, 90–100% of the WHC (high soil moisture availability—HSM), 50–60% of the WHC (moderate soil moisture availability–MSM), and 20–30% of the WHC (reduced soil moisture availability–RSM) were determined. For HSM treatment, plants (now 4–5 days old) in the first batch were allowed to dry down to 90% of the WHC before being watered back to 100%. For MSM treatment, plants in the second batch were allowed to dry down to 50% of the WHC before being watered back to 60% and then kept at this level until harvest. For RSM treatment, plants in the third batch were allowed to reach 20% of the WHC before being watered to 30% of the WHC and then kept at this level until harvested.

### 4.4. Impact of Soil Moisture Variations on Weed Susceptibility to Glyphosate

The experiments were conducted in the temperature-controlled glasshouse facility at the School of Agriculture and Food Sciences (SAFS), The University of Queensland (−27.51, 152.26), Gatton, during the summer of 2018–2019 and repeated in the same year. *C. truncata*, *S. oleraceus*, and *C. bonariensis* seedlings were established in each plastic pot (10-cm diameter) containing ca. 500 g of sun-dried soil collected from the UQ Crop Research Unit, The University of Queensland. At the 2-leaf stage, pots were allowed to continue growing under 3 soil moisture regimes as described above (HSM, MSM, and RSM). These 4 to 5-day old seedlings were grown and, later, maintained in a glasshouse with a mean temperature of 28/20 ± 2 °C, a photoperiod of 12/12 h (day/night), and a relative humidity of 65 ± 5%.

The plants of *C. truncata* were treated with glyphosate at the early tillering stage (~17–21 days after emergence), while *S. oleraceus* and *C. bonariensis* were treated at the rosette stage (~21–24 days after emergence and at the 6 to 8 leaf stage) with one of 7 concentrations of glyphosate [0, 62.5, 125, 250, 375, 500, and 750 g active ingredient (a.i.) ha^−1^]. The number of days to reach the abovementioned stage in HSM varied from 3 to 10 days for MSM and RSM. The glyphosate applied was WeedMaster DST^®^ salts (Nufarm, Victoria, Australia), and the original concentration of glyphosate (470 g L^−1^) was present as potassium and mono-ammonium. The application was undertaken in an enclosed cabinet track sprayer fitted with a Teejet nozzle (8003E) delivering 100.38 L ha^−1^ at 300 kPa. The herbicide was sprayed with the lowest rate of glyphosate first and finished with the highest rate. After herbicide application, the pots were transferred back within 30 min to their respective growing conditions in the temperature-controlled glasshouse.

Glyphosate sensitivity was recorded in 3 ways; viz. by a reduction in leaf chlorophyll content [as assessed by the SPAD meter (SPAD 502, Minolta Konica Ltd., Ramsey, New Jersey, USA)], by herbicide injury (%) at 21 days after glyphosate application (DAA), and by shoot dry biomass reduction at 21 DAA. For measuring the reduction in leaf chlorophyll content, SPAD values taken from treated plants and control plants were converted into a percent reduction in SPAD units, as an indication of declining chlorophyll content. In the second method, the injury level was visually assessed for herbicide symptoms, taking into account their visual appearance (e.g., tissue colour and necrosis, and other abnormalities) at 21 DAA using a modified scale proposed by Frans [61]. For measuring the biomass reduction at 21 DAA, the above-ground biomass of each biotype was determined by cutting shoots just above the soil surface, and oven-drying them for 72 h at 65 °C.

### 4.5. Impact of Soil Moisture Variations on Weed Morpho-Physiology and Growth

The seeds of *C. truncata*, *S. oleraceus*, and *C. bonariensis* were planted, and seedlings were raised in plastic pots (17 cm in diameter), containing 2.5 kg of heavy clay loam soil collected from the UQ Gatton campus, to study the effect of reduced soil moisture on weed growth. At the 3-leaf stage (ca. 4 to 5 days after seedling emergence), the seedlings were thinned to one uniform seedling per pot. From that point on, twelve pots per species were divided into 3 batches (HSM, MSM, and RSM) and monitored for growth and morphological differences. These 4 to 5-day-old seedlings were grown and, later, maintained in a glasshouse with a mean temperature of 28/20 ± 2 °C, a photoperiod of 12/12 h (day/night), and relative humidity of 65 ± 5%. The plant growth (viz. height attainment, number of leaves produced, leaf area developed, leaf thickness, chlorophyll content, stomatal conductance, and dry biomass) were all measured ca. 55 to 60 days after emergence (DAE), depending upon the weed species (see Peerzada [60] for details).

### 4.6. Experimental Designs and Statistical Analysis

Data from the 2 repeats of the study were pooled before analysis as the effect of the repeat was non-significant (*p* < 0.05) for most of the parameters measured. For the study on glyphosate efficacy, the experimental design was a 2-factor nested design including 7 glyphosate rates and 3 soil moisture regimes per weed species with 4 replications. The interval data on chlorophyll content was presented in line graphs to show the trend over time. Visual herbicide injury (%) was regressed over herbicide treatments using the 3-parameter sigmoid-sigmoidal model in Sigmaplot^®^ 14.5 (Systat Software, Inc., San Jose, CA, USA).
(1)$$\;y=\frac{a}{1+e^{-}{}^{\left(\frac{x-x_{0}}{b}\right)}}\;$$

In this regression model, *b* is the slope, *a* is the maximum visually assessed injury, *x* is the herbicide rate (g a.i. ha^−1^), and *x*_ο_ is the rate providing 50% response of the variable. Dry shoot biomass reduction was regressed over herbicide treatments using the 3-parameter logistic-sigmoidal model;
(2)$$y=\frac{a}{1+{\left(\frac{x}{x_{0}}\right)}^{b}}\;$$

In this regression model, *a* is denominated “saturation level”, corresponding to the bio-indicator response at the lower rate, *x*_ο_ is the inflection point of the curve, which corresponds to the *LD*_50_ value, and *b* describes the slope of the curve around *LD*_50_. Based on the analysis of the rate-response curves, the values of *LD*_50_ (rate required to cause 50% biomass reduction) with 95% confidence intervals (CI) and *LD*_80_ (rate required to cause 80% biomass reduction) values were estimated using Sigmaplot 14.0 and Microsoft® Excel 2016 (Microsoft Corporation, Redmond, Washington, USA). For individual weed species, the difference between qualitative (herbicide injury and biomass reduction) and quantitative variables (herbicide, rate, and soil moisture) were assessed by the CI. Overlapping CIs mean that there is no significant difference, whereas non-overlapping CIs indicate a significant difference. The estimated results of herbicide injury and biomass reduction of the weed species were also subjected to analysis of variance (ANOVA) using a general linear model (GLM) approach implemented in Minitab^®^ (Minitab Inc., State College, Pennsylvania, USA). To quantify the relative susceptibility of glyphosate under different soil moisture regimes, the ratio of tolerance to susceptibility, referred to as the tolerance factor (TF) index, was calculated (Equation (3)).
(3)$$TF=\frac{LD_{50}\;\left(RSM\right)\;or\;LD_{50}\;\left(MSM\right)\;}{LD_{50}\;\left(HSM\right)}\;$$

If the *TF* is not significantly different from 1.0, the *LD*_50_ is not different from unity, meaning there is no difference of efficacy in relation to the soil moisture regime.

For growth determination, the overall effect of soil moisture variations on the growth of the weed species was determined by means of a one-factor nested design including 3 soil moisture regimes per weed species with 4 replications, and treatment means were separated using Fisher’s protected *LSD* test at *p* < 0.05.

## 5. Conclusion and Future Directions

Results demonstrated that soil moisture-deficit stress induced more than two-fold glyphosate tolerance in tested weed species in comparison to the maximum soil moisture availability. Glyphosate susceptibility was considerably reduced in the control of tested weed species, especially at sublethal rates. The tolerance of glyphosate to sublethal rates could be related to decreased stomatal conductivity, increased thickness of leaves, high levels of chlorophyll in leaves, and decreased area of leaves. These changes under moderate and severe moisture-deficit stress could have interfered with the interception, retention, absorption, and translocation of glyphosate, but it did not affect the overall efficacy. Given the growing number of mild to severe forms of drought in the NGR, especially in the summer fallows, soil moisture levels at the time of application may interfere and reduce the efficacy of glyphosate when applied at recommended rates, depending upon the natural tolerance of weed biotypes to glyphosate. To avoid the efficacy reduction of glyphosate in the periods of drought, guidelines should be developed to improve weed control via manipulating application techniques, such as spray timing, use of adjuvants, different formulations, tank-mixing with contact herbicides, etc. which require further investigations.

The study also highlights that climatic variables, herbicide rate, and weed species, as well as their natural tolerance, interact to affect glyphosate tolerance. Therefore, understanding these relationships will ultimately provide better predictability of how a specific weed species and their ecotypes with differential natural herbicide tolerance (i.e., glyphosate tolerant and resistant biotypes), will respond to variations in soil moisture, either alone or in combination with other climatic conditions, such as temperature. Further studies involving different weed biotypes under varying temperatures and soil moisture regimes will help in the confirmation of this hypothesis. Better understanding of the interaction seen between climatic variables and a species’ natural tolerance would help to develop a set of guidelines for land managers in the application of glyphosate to manage weeds in the NGR.

## Figures and Tables

**Figure 1 plants-10-02345-f001:**
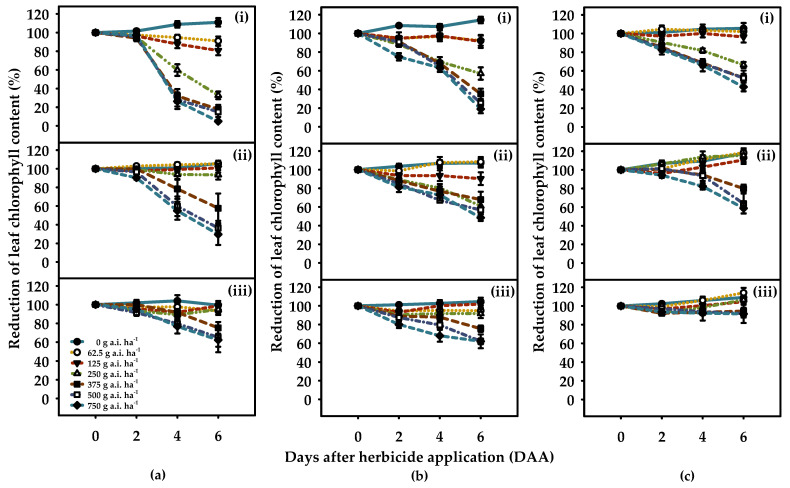
Reduction of leaf chlorophyll content, estimated in SPAD units, of glyphosate-treated *C. truncata* (**a**), *S. oleraceus* (**b**), and *C. bonariensis* (**c**) plants grown under (**i**) high soil moisture availability–HSM (90–100% of the WHC), (**ii**) moderate soil moisture availability–MSM (50–60% of the WHC) and (**iii**) reduced soil moisture availability–RSM (20–30% of the WHC), over the six days after the glyphosate application at various rates. Error bars represent the ± two standard errors of the mean (n = 8) from repeated experiments. The recommended rate for the control of *C. truncata*, *S. oleraceus*, and *C. bonariensis* in the Northern Grain Region’s summer fallow is 740, 705, and 658 g a.i. ha^−1^, respectively.

**Figure 2 plants-10-02345-f002:**
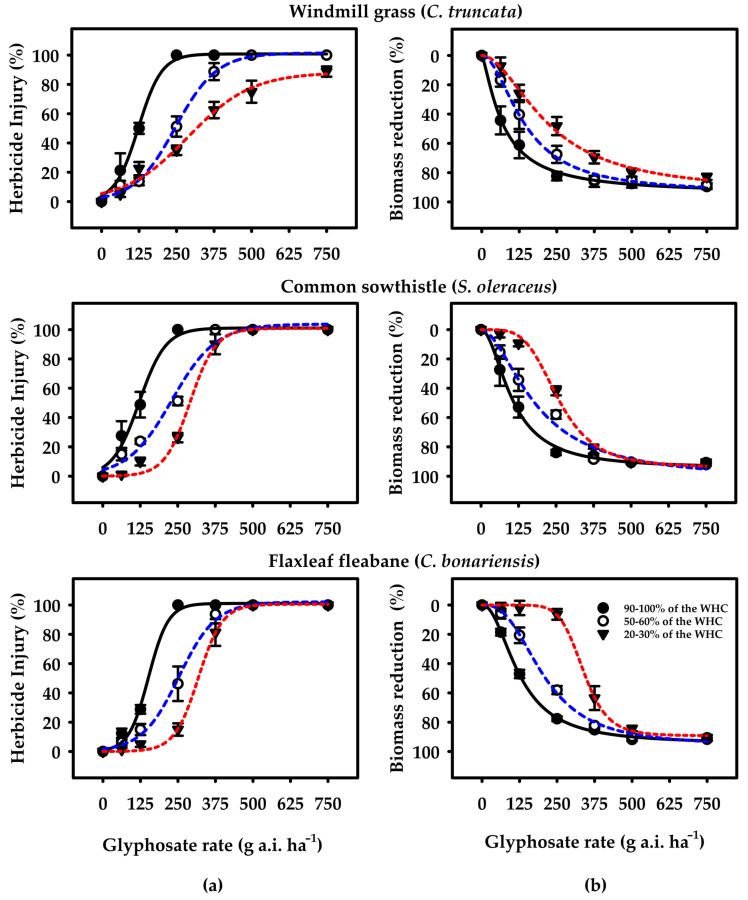
Herbicide control (**a**) and dry biomass reduction (**b**) of *C. truncata*, *S. oleraceus*, and *C. bonariensis* as a function of glyphosate rates (0, 63, 125, 250, 375, 500, and 750 g a.i. ha^−1^) and soil moisture regimes [(
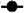
) 90–100% of the WHC = high soil moisture availability (HSM), (
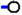
) 50–60% of the WHC = moderate soil moisture availability (MSM), and (
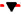
) 20–30% of the WHC = reduced soil moisture availability (RSM)] at 21 days after spraying. The figures show the observed data fitted to Equations (1) and (2). Error bars represent the ± two standard errors of the mean (n = 8) from repeated experiments.

**Table 1 plants-10-02345-t001:** Estimated parameter values [upper limit (a), slope (b), and LD_50_ with 95% confidence intervals (CI) and the tolerance factor (TF)] from the curve fittings using Equations (1) and (2) for the herbicide injury (%) and biomass reduction (%) of *C. truncata* grown under soil moisture regimes, measured at 21 days after treatment in response to glyphosate application at various rates. Standard error values are in parentheses.

Species	Parameters	Soil Moisture	a	b	LD_50_ *		TF **	LD_80_ **	R^2^
		% of the WHC	(%)		g a.i. ha^−1^	CI 95% ** Upper–Lower Limit		g a.i. ha^−1^	
*C. truncata*	Injury (%)	90–100%	100.8 ± 2.5	38.9 ± 7.0	122.2	115.7–128.6	1.0	175.2	0.91
		50–60%	101.5 ± 2.8	68.8 ± 7.7	244.5	235.2–253.7	2.0	337.0	0.94
		20–30%	88.2 ± 4.6	105.8 ± 14.9	316.7	297.0–336.4	2.6	528.7	0.87
	Biomass reduction (%)	90–100%	94.6 ± 8.4	−1.3 ± 0.5	77.8	66.2–89.4	1.0	236.1	0.81
		50–60%	93.8 ± 6.3	−1.9 ± 0.4	153.9	137.5–170.3	2.0	338.8	0.87
		20–30%	93.7 ± 9.1	−1.8 ± 0.4	239.4	208.5–270.3	2.6	506.0	0.87
*S. oleraceus*	Injury (%)	90–100%	101.1 ± 2.8	44.1 ± 8.1	120.7	112.8–128.5	1.0	180.4	0.88
		50–60%	103.8 ± 2.2	73.7 ± 5.8	225.3	218.1–232.6	1.9	320.2	0.96
		20–30%	100.9 ± 2.4	42.7 ± 5.3	290.4	283.7–297.0	2.4	348.5	0.96
	Biomass reduction (%)	90–100%	94.7 ± 5.1	−1.9 ± 0.4	110.1	99.5–120.7	1.0	234.6	0.85
		50–60%	102.7 ± 6.9	−1.8 ± 0.3	178.9	159.4–198.4	1.6	340.6	0.91
		20–30%	94.1 ± 2.0	−4.2 ± 0.4	264.9	259.7–270.1	2.4	373.9	0.98
*C. bonariensis*	Injury (%)	90–100%	101.1 ± 1.1	32.9 ± 3.2	150.9	147.2–154.6	1.0	195.6	0.98
		50–60%	102.1 ± 3.9	61.6 ± 10.6	250.9	239.0–262.7	1.7	332.7	0.89
		20–30%	100.6 ± 2.8	39.9 ± 5.1	317.6	309.8–325.4	2.1	372.3	0.94
	Biomass reduction (%)	90–100%	94.9 ± 1.8	−2.1 ± 0.1	131.1	126.9–135.3	1.0	289.0	0.98
		50–60%	96.1 ± 3.3	−2.7 ± 0.3	211.6	201.9–221.3	1.6	363.1	0.95
		20–30%	89.3 ± 3.3	−8.3 ± 1.6	347.2	343.2–361.6	2.6	474.2	0.92

LD_50_ is the glyphosate rate (g a.i. ha^−1^) required to cause 50% injury or biomass reduction; LD_80_ is the glyphosate rate (g a.i. ha^−1^) required to cause 80% injury or biomass reduction; R^2^ is the conservative coefficient of determination; CI 95% = 95% confidence interval; TF = Tolerance factor; 90–100% of the WHC = high soil moisture availability (HSM); 50–60% of the WHC = moderate soil moisture availability (MSM); 20–30% of the WHC = reduced soil moisture availability (RSM). * Values adjusted by Sigmaplot program. ** Estimated index through the Excel spreadsheet.

**Table 2 plants-10-02345-t002:** Influence of soil moisture regimes on the morpho-physiology and growth of *C. truncata*, measured at 55 to 60 days after emergence. Data collected from repeated experiments with a total of eight replications per treatment.

Soil Moisture	Plant Height	No. of Leaves	Chlorophyll Content	Stomatal Conductance	Leaf Thickness	Leaf Area	Dry Weight
% of the WHC	cm	-	SPAD Units	mmol m^−2^ s^−1^	µm	cm^2^	g
90–100%	47.8 ± 2.5	94 ± 3.4	28.2 ± 0.4	163.8 ± 5.2	95.8 ± 2.1	267.1 ± 6.1	3.3 ± 0.1
50–60%	35.3 ± 1.3	71 ± 3.0	36.2 ± 0.5	127.4 ± 4.8	105.8 ± 1.9	174.2 ± 4.5	2.2 ± 0.1
20–30%	25.8 ± 1.7	47 ± 2.6	46.2 ± 0.5	80.6 ± 2.4	114.6 ± 1.2	129.7 ± 3.6	1.4 ± 0.05
*p*-value	<0.001	<0.001	<0.001	<0.001	<0.001	<0.001	<0.001
LSD (*p* < 0.05)	5.7	8.8	1.3	12.6	5.3	14.2	0.3

90–100% of the WHC = high soil moisture availability (HSM); 50–60% of the WHC = moderate soil moisture availability (MSM); 20–30% of the WHC = reduced soil moisture availability (RSM); LSD = Least Significant Difference.

**Table 3 plants-10-02345-t003:** Influence of soil moisture regimes on the morpho-physiology and growth of *S. oleraceus*, measured at 55 to 60 days after emergence. Data collected from repeated experiments with a total of eight replications per treatment.

Soil Moisture	Plant Height	No. of Leaves	Chlorophyll Content	Stomatal Conductance	Leaf Thickness	Leaf Area	Dry Weight
% of the WHC	cm	-	SPAD Units	mmol m^−2^ s^−1^	µm	cm^2^	g
90–100%	64.8 ± 1.9	27.6 ± 1.0	36.0 ± 1.1	288.9 ± 6.1	308.5 ± 5.1	423.5 ± 15.5	3.1 ± 0.06
50–60%	49.6 ± 1.4	18.0 ± 0.9	42.1 ± 1.3	211.5 ± 4.4	327.4 ± 6.0	245.5 ± 14.0	2.1 ± 0.09
20–30%	20.5 ± 0.8	14.5 ± 1.1	47.2 ± 1.1	108.2 ± 5.1	371.2 ± 9.3	166.2 ± 8.7	1.2 ± 0.03
*p*-value	<0.001	<0.001	<0.001	<0.001	<0.001	<0.001	<0.001
LSD (*p* < 0.05)	4.3	2.9	3.4	15.5	20.72	38.4	0.2

90–100% of the WHC = high soil moisture availability (HSM); 50–60% of the WHC = moderate soil moisture availability (MSM); 20–30% of the WHC = reduced soil moisture availability (RSM); LSD = Least Significant Difference.

**Table 4 plants-10-02345-t004:** Influence of soil moisture regimes on the morpho-physiology and growth of *C. bonariensis*, measured at 55 to 60 days after emergence. Data collected from repeated experiments with a total of eight replications per treatment.

Soil Moisture	Plant Height	No. of Leaves	Chlorophyll Content	Stomatal Conductance	Leaf Thickness	Leaf Area	Dry Weight
% of the WHC	cm	-	SPAD Units	mmol m^−2^ s^−1^	µm	cm^2^	g
90–100%	39.3 ± 1.2	97.8 ± 2.5	35.6 ± 2.0	360.5 ± 11.4	293.0 ± 8.3	250.2 ± 4.1	2.8 ± 0.06
50–60%	16.5 ± 0.9	55.6 ± 2.3	41.5 ± 1.4	188.8 ± 9.0	335.4 ± 9.2	160.9 ± 3.9	2.1 ± 0.05
20–30%	12.6 ± 0.2	25.9 ± 0.5	50.2 ± 1.5	96.2 ± 6.8	367.7 ± 7.8	87.7 ± 2.9	0.9 ± 0.02
*p*-value	<0.001	<0.001	<0.001	<0.001	<0.001	<0.001	<0.001
LSD (*p* < 0.05)	3.4	5.8	4.9	27.3	24.8	10.9	0.2

90–100% of the WHC = high soil moisture availability (HSM); 50–60% of the WHC = moderate soil moisture availability (MSM); 20–30% of the WHC = reduced soil moisture availability (RSM); LSD = Least Significant Difference.

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
