# Peer review of "Effect of Soil Moisture Regimes on the Glyphosate Sensitivity and Morpho-Physiological Traits of Windmill Grass (Chloris truncata R.Br.), Common Sowthistle (Sonchus oleraceus L.), and Flaxleaf Fleabane [Conyza bonariensis (L.) Cronq.]"

_plants, 2021, doi:10.3390/plants10112345_

Round 1
Reviewer 1 Report
This is quite interesting and very topical paper that compares glyphosate efficiency on three species of horrible weeds in Australia under three scenarios of water supply. The findings are interesting and providing further support for the notion that drought conditions might alter weed response to herbicide applications in the field. The experimental work was well conducted and the manuscript is well written. My major concern relates to the clarity of presentation, especially in sections about statistical analysis, and also regarding to figures and tables. My comments are listed below.
l 40 - phosphate
l 45 - NGR - the meaning of this abbreviation needs to be defined again in the main text. Imagine that one starts reading by skipping the abstract.
ll 86-88 - I am getting puzzled here - in case sensitivity of weeds increases then the efficacy of the herbicide would increase too, right?
Results - here I suggest merging figures and tables that show the same thing but for different species of weeds; e.g. merge Figures 1-3 in one figure with nine panels. In this way you would not need to repeat the figure and table captions over and over, and the main text would be easier to read. Also the paper would look nicer from an aesthetic point of view, I think. Some figures, especially Figures 1-3, are of low quality and are difficult to read. Please enlarge and change the font size. There are no p-values related to the main message of the paper; this is strange since one of the conclusions is that there are species-specific differences in response to glyphosate application under different watering regimes, which now is only supported by visual impression from the figures (and text description, of course); statistical support is needed. In tables, new coding for water treatments are introduced, which is redundant. Stick to one system, please.
l 135 - what is the tolerance factor? This parameter needs to be explained in the methods.
l 441-2 - I think you can omit this sentence as it does not bring anything new to the story.
section 4.6 - this is totally unclear. The convention is that ANOVA stands for analysis of variance, which is different approach to GLM. Using both terms in one sentence makes me wonder if the entire analysis was made correctly. More details are needed. In the same sentence it is not clear to which data it is related. The non-linear regressions seem to be OK, but how did you compare the curves in your two-factor nested design? Certainly not by using ANOVA. I understand the last sentence, which seems to be alright given that normality of the residuals was assured (is it correct that the calculations are based only on 4 plants per category?).
I guest the Data availability statement can be removed entirely?
My last comment goes more to the journal rather than the authors, but practice of moving the method sections is impracticable and makes reading (and evaluating) the article more difficult.
Author Response
My comments are listed below.
l 40 - phosphate
Response: addressed.
l 45 - NGR - the meaning of this abbreviation needs to be defined again in the main text. Imagine that one starts reading by skipping the abstract.
Response: addressed.
ll 86-88 - I am getting puzzled here - in case sensitivity of weeds increases then the efficacy of the herbicide would increase too, right?
Response: Yes. corrected.
Results - here I suggest merging figures and tables that show the same thing but for different species of weeds; e.g. merge Figures 1-3 in one figure with nine panels. In this way you would not need to repeat the figure and table captions over and over, and the main text would be easier to read. Also the paper would look nicer from an aesthetic point of view, I think. Some figures, especially Figures 1-3, are of low quality and are difficult to read. Please enlarge and change the font size. There are no p-values related to the main message of the paper; this is strange since one of the conclusions is that there are species-specific differences in response to glyphosate application under different watering regimes, which now is only supported by visual impression from the figures (and text description, of course); statistical support is needed. In tables, new coding for water treatments are introduced, which is redundant. Stick to one system, please.
Response: Figures has been edited as per instruction
l 135 - what is the tolerance factor? This parameter needs to be explained in the methods.
Response: Equation has been included.
l 441-2 - I think you can omit this sentence as it does not bring anything new to the story.
Response: it refers to the complete process. It would be helpful if anyone wants to read the whole method of developing soil moisture levels.
section 4.6 - this is totally unclear. The convention is that ANOVA stands for analysis of variance, which is different approach to GLM. Using both terms in one sentence makes me wonder if the entire analysis was made correctly. More details are needed. In the same sentence it is not clear to which data it is related. The non-linear regressions seem to be OK, but how did you compare the curves in your two-factor nested design? Certainly not by using ANOVA. I understand the last sentence, which seems to be alright given that normality of the residuals was assured (is it correct that the calculations are based only on 4 plants per category?).
Response: The statistical analysis has been simplified.
I guest the Data availability statement can be removed entirely?
Response: Removed.
My last comment goes more to the journal rather than the authors, but practice of moving the method sections is impracticable and makes reading (and evaluating) the article more difficult.
Response: Yes, it makes it hard for the reader to understand the results.
Reviewer 2 Report
The paper under review deals with the research on the effect of soil moisture variations in the control efficiency of glyphosate on C. truncata, S. oleraceus, and C. bonariensis (ii) to evaluate the tolerance of 16 tested weed species under water stresses, and (iii) to determine the morphological and physiological characteristics of these species to partially explain herbicide tolerance under soil moisture variations. The article is not properly structured. After the introduction, the results were presented, followed by a discussion, and only then the research methodology. In my opinion, the novelty of the research is debatable. I think the Authors should indicate the novelty. The very legitimacy of the research undertaken raises my doubts. The same results could be predicted without experimental studies, e.g. the negative effect of moisture stress on weed growth and structure. Some conclusions result from the research carried out by other authors. The paper is a research report, not a research article. The aim of the experimental studies is not sufficiently justified. The introduction is too general. There are no newer (after 2010) literature items in the bibliography. The font in the drawings is not sufficiently visible. The signature under Figure 1 should be corrected in accordance with the form.
Author Response
The paper under review deals with the research on the effect of soil moisture variations in the control efficiency of glyphosate on C. truncata, S. oleraceus, and C. bonariensis (ii) to evaluate the tolerance of 16 tested weed species under water stresses, and (iii) to determine the morphological and physiological characteristics of these species to partially explain herbicide tolerance under soil moisture variations.
The article is not properly structured. After the introduction, the results were presented, followed by a discussion, and only then the research methodology.
Response: I agree with you. Authors are bound to follow the journal’s style.
In my opinion, the novelty of the research is debatable. I think the Authors should indicate the novelty.
Response: Previously, studies on glyphosate efficacy under stress conditions focused on half, single, or double of the recommended rate. However. our research focuses on climate-induced herbicide tolerance in a susceptible biotype. The study will highlight the importance of sub-lethal herbicide rate and climate change in the slow evolution of herbicide resistance, especially non-targeted site resistance.
The very legitimacy of the research undertaken raises my doubts. The same results could be predicted without experimental studies, e.g. the negative effect of moisture stress on weed growth and structure.
Response: If we look at this through the principle of weed evolution, every single weed species and their population experience different selection pressures (i.e., absence or presence of herbicide, high temperature, drought, and other cultural practices, etc.). Thus, responses to stresses (i.e., herbicides or environmental factors) over generations may vary within weed species and even within a weed population. For example, when I subjected S. oleraceus, a susceptible biotype, to elevated temperature with reduced soil moisture, glyphosate efficacy improved (not published). Sometimes, our predictions about herbicide efficacy can go wrong.
Some conclusions result from the research carried out by other authors.
Response: improved.
The paper is a research report, not a research article. The aim of the experimental studies is not sufficiently justified.
The introduction is too general.
Response: Has been added.
There are no newer (after 2010) literature items in the bibliography.
Response: We used old references for some statements which can not be denied by recent research improvements. Authors are strictly against secondary citations.
The font in the drawings is not sufficiently visible. The signature under Figure 1 should be corrected in accordance with the form.
Response: Corrected
Reviewer 3 Report
The main doubt of the reviewer regarding the publication of the manuscript on “Effect of soil moisture variations on the growth and glyphosate sensitivity of windmill grass, common sowthistle, and flaxleaf fleabane” is the fact that water deficit alone has also a negative effect on morpho-physiological features of weeds, even without glyphosate, so it difficult to divide these effects.
The order in which the results are presented in Results section does not follow the order of the aims in the Introduction section. In the purposes of the study, first, the impact of the water deficit in the soil itself on the physico-chemical characteristics of selected weed species was presented, and then together with glyphosate, but the results in Results chapter are discussed in reverse order, which disturbs the sequence of thought and deduction of reader.
The second doubt of the reviewer is that the reaction of weeds could be different under the conditions of the pot experiment than under the conditions in the field. Moreover, it is difficult to obtain appropriate humidity conditions in pots with a diameter of 10 cm.
Detailed comments:
Title: Latin names for weeds should be added.
Abstract:
The term „moisture stress” should be explained at the begining. Is it water deficiency or water excess?
The abbreviations first time used in the text should be explained.
Results:
Very detailed description of the results, even too much detailed in some parts.
Different symbols are used in the tables and figures than in the main text so it's difficult to read and understand. Please standardize these symbols for the levels of the moisture stress factor.
Sometimes there are mistakes in the symbols, e.g. lines 164-168.
M&M:
How many plants were examined? How many replications?
Conclusions:
Conclusions should be changed/improved.
Some remarks were put in the pdf of the paper.

Author Response
The main doubt of the reviewer regarding the publication of the manuscript on “Effect of soil moisture variations on the growth and glyphosate sensitivity of windmill grass, common sowthistle, and flaxleaf fleabane” is the fact that water deficit alone has also a negative effect on morpho-physiological features of weeds, even without glyphosate, so it difficult to divide these effects.
The order in which the results are presented in Results section does not follow the order of the aims in the Introduction section. In the purposes of the study, first, the impact of the water deficit in the soil itself on the physico-chemical characteristics of selected weed species was presented, and then together with glyphosate, but the results in Results chapter are discussed in reverse order, which disturbs the sequence of thought and deduction of reader.
Response: Corrected.
The second doubt of the reviewer is that the reaction of weeds could be different under the conditions of the pot experiment than under the conditions in the field. Moreover, it is difficult to obtain appropriate humidity conditions in pots with a diameter of 10 cm.
Response: The study focuses on the water-deficit stress-induced herbicide tolerance in susceptible biotypes. It has been mentioned that results can vary under field conditions, and more research is required to integrate climatic variables with more weed biotypes to predict herbicide efficacy in the field conditions.
Detailed comments:
Title: Latin names for weeds should be added.
Response: Added
Abstract:
The term „moisture stress” should be explained at the begining. Is it water deficiency or water excess?
Response: corrected.
The abbreviations first time used in the text should be explained.
Response: Abbreviations has been corrected.
Results:
Very detailed description of the results, even too much detailed in some parts.
Response: Edited.
Different symbols are used in the tables and figures than in the main text so it's difficult to read and understand. Please standardize these symbols for the levels of the moisture stress factor.
Response: Corrected
Sometimes there are mistakes in the symbols, e.g. lines 164-168.
Response: Mistakes have been corrected.
M&M:
How many plants were examined? How many replications?
Response: Added in the M&M.
Conclusions:
Conclusions should be changed/improved.
Response: Improved.
Some remarks were put in the pdf of the paper.
Response: Addressed all comments
Reviewer 4 Report
Overall: This manuscript looks at climatic / environmental impacts on plant phenotype and glyphosate tolerance. I believe this to be very important and understudied area of research,. The authors include both visual estimates of damage (herbicide control) and a quantitative measure (biomass), which is excellent. The manuscript is well written and the experimental design and analysis appear correct. They conducted regression based on a positive response in their experimental design analysis (ANOVA). My only concern is that results are reproduced in time and space. I suspect they are but please clarify.
L24: “… growth with reduced soil moisture…” to “growth. Reduced soil moisture…”
L70 to 71: Please clarify spray droplet behavior.
Intro overall: A good level of detail. I would recommend including just a brief overview of the weeds habit for context to the reviewers. Perhaps some insight into why they are damaging would give additional context.
L123: Your formatting for the Figure 1 caption cuts oddly here and ends on 124, which is below your Figure 3 caption.
L172 to 173: Excellent observation on regrowth.
L426: Was this repeated?
L510: reading through, I couldn’t tell if the experiments were repeated in time and space to demonstrate reproducibility of the results. I saw the mention of treatment by experiment interaction (L515) so please clarify if so and how many times.
Figure 1 – Please use the abbreviations from the text in the figure caption as well, for clarity and continuity.
L526: Please indicate fit statistics for your regression models. You used R2, how was this calculated, via SigmaPlot or by hand?
Discussion: I felt the discussion was adequate around direct result findings but I would recommend inclusion of how your findings may impact production systems in your environment that are using glyphosate to manage these weeds? Should alternative or sequential applications be used in drought years? It’s mentioned in the abstract that this work occurs relative to the grain region in Australia. How do your results impact the grain industry in Australia?
Author Response
L24: “… growth with reduced soil moisture…” to “growth. Reduced soil moisture…”
Response: corrected.
L70 to 71: Please clarify spray droplet behavior.
Response: sentence edited.
Intro overall: A good level of detail. I would recommend including just a brief overview of the weeds habit for context to the reviewers. Perhaps some insight into why they are damaging would give additional context.
Response: included.
L123: Your formatting for the Figure 1 caption cuts oddly here and ends on 124, which is below your Figure 3 caption.
Response: corrected.
L172 to 173: Excellent observation on regrowth.
L426: Was this repeated?
Response: Yes.
L510: reading through, I couldn’t tell if the experiments were repeated in time and space to demonstrate reproducibility of the results. I saw the mention of treatment by experiment interaction (L515) so please clarify if so and how many times.
Response: Edited.
Figure 1 – Please use the abbreviations from the text in the figure caption as well, for clarity and continuity.
Response: Corrected.
L526: Please indicate fit statistics for your regression models. You used R2, how was this calculated, via SigmaPlot or by hand?
Response: It was calculated via Sigmaplot.
Discussion: I felt the discussion was adequate around direct result findings but I would recommend inclusion of how your findings may impact production systems in your environment that are using glyphosate to manage these weeds? Should alternative or sequential applications be used in drought years? It’s mentioned in the abstract that this work occurs relative to the grain region in Australia. How do your results impact the grain industry in Australia?
Response: Conclusion has been edited as per advice.
Round 2
Reviewer 1 Report
Review of the revised version of the paper by Peerzada et al., submitted to Plants
I see numerous changes made to the text by the authors after the first round of reviews. I appreciate that the changes made are showed by different colour so these can be easily tracked. However, I the responses to most of the review queries lack details, which is not acceptable. I require further changes to be made to the manuscript before this paper can be accepted.
In the first place the new figures are still of very low quality. Please make sure that figures you are embedding to the text are good enough so all the lines and letters are distinguishable and readible.
Importantly, the authors failed to notice my suggestion about merging the figures and tables. In my oppinion doing so the paper would be easier to read since now it is broken down to small fragments of the text by too many tables and figures, which have the same headings. Merging the corresponding tables (e.g. Tables 1, 2 and 3, etc.) and figures would therefore shorten the paper a bit.
More care needs to be done when preparing the tables. For example, in Tables 1-3 the heading for the first collumn is "Species" but does not contain species information.
In figures showing the relationship of the herbicide injury with glyphosate dose, the curves clearly exceed 100 %, which is not possible. I suspect that these curves are not based on any rigorous analysis, and the level of my doubts about the correctness of the analysis has further increased. Also, the text regarding the analysis is not clear enough, and does not explain enough what has been (or should have been) done. I require thorough explanation and description of the curve fitting process, together with providing any indication of the significance of the curves shown, and significance of the difference between the curves of different treatments. I could not find this in the paper. Additionally, why those particular curves were chosen? I believe that due to the nature of the data you could have used binomial GLM, which is a robust approach, allows for comparison between the curves and also provides all the necessary statistics, inclusing assymetric 95% CI. If the chosen curves have some ground in the theory or in the literature, it should be cited.
section 4.6 first sentence - I think there are too many verbs
Author Response
Response to Reviewer 1 Comments
Point 1: I see numerous changes made to the text by the authors after the first round of reviews. I appreciate that the changes made are showed by different colour so these can be easily tracked. However, I the responses to most of the review queries lack details, which is not acceptable. I require further changes to be made to the manuscript before this paper can be accepted.
Response 1: I apologies, if I missed something or failed to addressed the comments properly.
Point 2: In the first place the new figures are still of very low quality. Please make sure that figures you are embedding to the text are good enough so all the lines and letters are distinguishable and readible.
Response 2: The quality of the figures has been improved.
Point 3: Importantly, the authors failed to notice my suggestion about merging the figures and tables. In my oppinion doing so the paper would be easier to read since now it is broken down to small fragments of the text by too many tables and figures, which have the same headings. Merging the corresponding tables (e.g. Tables 1, 2 and 3, etc.) and figures would therefore shorten the paper a bit.
Response 3: The figures and tables have been merged.
Point 4: More care needs to be done when preparing the tables. For example, in Tables 1-3 the heading for the first collumn is "Species" but does not contain species information.
Response 4: Tables have been edited.
Point 5: In figures showing the relationship of the herbicide injury with glyphosate dose, the curves clearly exceed 100 %, which is not possible. I suspect that these curves are not based on any rigorous analysis, and the level of my doubts about the correctness of the analysis has further increased. Also, the text regarding the analysis is not clear enough, and does not explain enough what has been (or should have been) done. I require thorough explanation and description of the curve fitting process, together with providing any indication of the significance of the curves shown, and significance of the difference between the curves of different treatments. I could not find this in the paper. Additionally, why those particular curves were chosen? I believe that due to the nature of the data you could have used binomial GLM, which is a robust approach, allows for comparison between the curves and also provides all the necessary statistics, inclusing assymetric 95% CI. If the chosen curves have some ground in the theory or in the literature, it should be cited.
Response 5: In these experiments, a weed species was exposed to a herbicide and the response to herbicide injury or biomass reduction was measured. The analysis consisted of plotting the herbicide rate vs. herbicide injury or biomass reduction and applying a non-linear regression model. The result is an upward sigmoidal curve (Herbicide injury) and downward sigmoidal curve (Biomass reduction). For individual weed species, the difference between qualitative (herbicide injury and biomass reduction) and quantitative variables (herbicide, rate, and soil moisture) were assessed by the CI. Overlapping CIs mean that there is no significant difference, whereas non-overlapping CIs indicate a significant difference. Three-parameter sigmoid-sigmoidal and sigmoid-logistic models are usually used for herbicide injury and biomass reduction, respectively. The means of the factors were also compared by using GLM in Minitab.
Point 7: section 4.6 first sentence - I think there are too many verbs
Response 7: The section has been edited.
Reviewer 2 Report
I appreciate the changes introduced by the Authors and the explanations provided. However, the structure of the article must be changed: introduction, research methodology, results, discussion, and conclusions.
Author Response
Thank you for your kind contribution, which helped us in improving the quality of this manuscript. I really appreciate this. I completely agree with your suggestion regarding the structure of the manuscript but this is the style proposed by the journal to keep M&M at the end.
Reviewer 3 Report
Authors improved the manuscript to be more clear. But I still have serious remarks according to the concept od the experiment. Among treated weed plants can be genetically resistant to glyphosate because event very high doses of active ingredient (750 g a.i.) did not affect some weed Species. Herbicide resistant biotype of weeds are more and more often especially in in regions, where the same herbicide was used, for example glyphosate. In such cases it doesn't depend on soil moisture level. Moreover, I do not agreement with the conclusion on the last lentence od Abstrakt. It is difficult to say if morpho-physiological traits caused by water deficiency influence the herbicide sensitivity it depend on herbicide absorption and transport or rather herbicide caused morpho-physiological changes in tested plants.
Conclusion section is too long. First sentence needs correction. Conclusions do not fully relate to the conducted research. It contains literature citation.
Author Response
Among treated weed plants can be genetically resistant to glyphosate because event very high doses of active ingredients (750 g a.i.) did not affect some weed Species. Herbicide-resistant biotypes of weeds are more and more often especially in regions, where the same herbicide was used, for example, glyphosate. In such cases, it doesn't depend on soil moisture level.
Ans: I completely agree with you. Yes, the herbicide has been widely used in the region, but over aim was to set the story that climate change can be a factor that exacerbates herbicide resistance through recurrent selection. That's why susceptible biotype has been used in this study. As you might know that the northern grain region of Australia experiences severe drought in summer periods, and the degree of moisture stress varies with weed species (so weeds might not show any stress symptoms), which could possibly be the reason that some growers might confuse climate-induced herbicide tolerance with resistance.
Moreover, I do not agree with the conclusion in the last sentence of the Abstract. It is difficult to say if morpho-physiological traits caused by water deficiency influence the herbicide sensitivity it depends on herbicide absorption and transport or rather an herbicide caused morpho-physiological changes in tested plants.
Ans: Here, I disagree with you. To understand herbicide efficacy, it is very important to understand the herbicide mechanism of action and mode of herbicide tolerance in plants. If we talk about non-targeted site mechanisms of herbicide resistance, such as reduced absorption or translocation, vacuolar sequestration, or herbicide metabolism, and even targeted site (over-expression or excessive production of target site), they can easily be regulated by environmental stress. In elevated temperature, I observed rapid reduction of leaf chlorophyll content at older leaves at sub-lethal herbicide rate but later regrowth of lateral shoots was observed, which is the clear indication that herbicide is not reaching the apoplast (it can be vacuolar sequestration). Similarly, in reduced soil moisture, plants try to curl leaves ( in case of windmill grass) or develop shorter leaves (Fleabane and sowthistle), which, reduces herbicide interception; slight erect leaf orientation was also noticed for fleabane and sowthistle in reduced soil moisture conditions (limited interception). Higher leaf chlorophyll content was observed under stress conditions because the plant was producing thicker leaves, which could have increased the number of sites of action (In herbicide-tolerant crops, this mechanism is used). So, there could be a direct connection between changes in leaf morpho-physiological characteristics and the efficacy of most foliar-applied herbicides.
The conclusion section is too long. The first sentence needs correction. Conclusions do not fully relate to the conducted research. It contains literature citations.
Ans: Conclusion has been edited.